# Implementation and Baseline Evaluation of an Evidence-Based Group Antenatal Care Program in Two Nigerian States

**DOI:** 10.3390/ijerph21111461

**Published:** 2024-10-31

**Authors:** William Douglas Evans, Chinwe L. Ochu, Jeffery B. Bingenheimer, Samson Babatunde Adebayo, Fasiku Adekunle David, Sani Ali Gar, Masduk Abdulkarim

**Affiliations:** 1Milken Institute School of Public Health, The George Washington University, 950 New Hampshire Avenue NW, Washington, DC 20052, USA; chinwelucia.ochu@gwmail.gwu.edu (C.L.O.); bartbing@gwu.edu (J.B.B.); 2Data Research and Mapping Consults, Suite: B 14, Dansarari Plaza, 5 Ziguinchor Str, Wuse Zone 4, Abuja 900284, Nigeria; adebayosam@hotmail.com (S.B.A.); fashadetoyin@yahoo.com (F.A.D.); saligar58@yahoo.com (S.A.G.); 3Bill & Melinda Gates Foundation, Nigeria Country Office, Global Development Division, 45 Aguiyi Ironsi St, Wuse, Abuja 904101, Nigeria; masduk.abdulkarim@gatesfoundation.org

**Keywords:** group antenatal care, perinatal care, family planning, maternal and child health, program evaluation, implementation science, low- and middle-income countries, Nigeria

## Abstract

Northern Nigeria has had historically low antenatal care (ANC) utilization rates with poor health outcomes. Previous studies have shown that group antenatal care (gANC) improves ANC behavior and pregnancy outcomes. The gANC has been adopted in Kaduna and Kano States, Nigeria. This paper presents baseline findings from the implementation of the gANC program in Kaduna and Kano States, Nigeria, based on data collected from 1269 and 1200 pregnant women, respectively, from March to April 2024. Analyses of sociodemographic and pregnancy behavior data were performed. Participants were mostly between the age of 19 and 31 years, married or living with a partner, with over 50% having their own businesses. Over 62% and 34% had completed secondary- or higher-level education, with 60% and 80% living in urban areas in Kaduna and Kano States, respectively. In Kano State, >60% of the women had their last delivery at home, with 41.6% not assisted by a skilled birth attendant. In Kaduna, >63% delivered in the hospital and >50% had skilled attendance during labor. Almost half had not used contraceptives previously. This study has provided baseline evaluation data for the implementation of gANC in two states in Nigeria. Subsequent longitudinal data will examine the impact of gANC utilization on perinatal outcomes and contraceptive behavior to inform the scaling of the program in the country.

## 1. Introduction

Nigeria has historically low rates of antenatal care (ANC) utilization, with pronounced disparities in rural areas and the northwest region [1,2]. Analysis of the Nigeria 2018 Demographic and Health Survey (DHS) showed only 67% ANC use among women aged 15–49 years, with some states in the northwest having a non-utilization rate of over 50% [2]. This was more pronounced in rural areas with a non-utilization rate of 33.8% compared to 10.1% in urban areas. Facility births were lowest in the northwest, at 16%, compared to 82% in the southeast [2,3]. Having no contact with ANC has been observed in up to 25% of women in Nigeria [3]. Literacy level, living distance to health facility, healthcare experience, and gender relations are among the factors that influence utilization of ANC and contraceptive uptake [4].

Previous studies have demonstrated the effectiveness of group antenatal care (gANC) in low- and middle-income countries (LMICs). Group ANC is a concept that enables peer-to-peer discussions and sharing of experiences [5,6]. This approach seeks to address quality of care and coverage gaps prevalent in conventional individual person–doctor antenatal sessions driven by poor communications and limited counselling time [5]. Women participating in gANC were more likely to discuss reproductive health issues, including family planning and HIV testing, with their partners compared to those in conventional ANC in a randomized controlled trial in Malawi and Tanzania [6]. In a pre–post quasi-experimental study of over 1600 women in Kenya, knowledge of three or more danger signals of pregnancy rose to 26% following completion of gANC, more than three times the 7.1% observed at baseline [7]. This was also true for many other quality-of-care parameters. In a study in Nigeria and Kenya, women receiving gANC were more likely than their counterparts in individual care to attend no less than four ANC visits (Nigeria: aOR 13.30, CI 7.69–22.99, *p* < 0.001; Kenya: aOR 7.12, CI 3.91–12.97, *p* < 0.001), receive good-quality care (Nigeria: aOR 5.8, CI 1.98–17.21, *p* < 0.001; Kenya: aOR 5.08, CI 2.31–11.16, *p* < 0.001), and deliver in a health facility (Nigeria: aOR 2.30, CI 1.51–3.49) [8]. In another study, Senegalese adolescents attending gANC indicated preference for utilization of gANC over individual ANC in their next pregnancies [9]. Open communication and supportive relationships have also been associated with gANC [9].

Failure to detect a significant difference for the impact of gANC on pregnancy outcomes compared to individual ANC has also been described in the literature. In a cluster-randomized controlled trial to determine the effect of gANC on gestational length in a large sample of pregnant women (4752 intervention, 4091 control) in Rwanda, there was no significant difference in gestational length or other secondary outcomes, except for postnatal visits, which were more common in the control group [10]. The authors noted that low dose of exposure to the intervention at only four antenatal visits might have influenced this observation [10]. In the qualitative component of the same trial, participants of gANC reported knowledge gain, support from peers, and improved relationship with their healthcare providers following the intervention [10]. However, these were pre- and post-test data from a study that did not include a comparison group.

The World Health Organization (WHO) recommends group antenatal care (gANC) as an effective model to improve maternal and newborn health outcomes [11]. In Nigeria, the Federal Ministry of Health (FMOH) has endorsed gANC as a strategy to increase the quality and coverage of ANC services. However, the adoption and adaptation of gANC in Nigeria has been limited. This study seeks to demonstrate the effectiveness of antenatal care (ANC)/postnatal care (PNC) packages and scale proven implementation packages; this has only been performed in a limited number of countries. It will catalyze scaling platforms to advance primary healthcare (PHC) as well as scale contextual proven intervention BOWs. This study will help answer questions about the effectiveness of gANC in different settings as well as how gANC can be designed to be successful in different settings [12]. It will provide governments of participating states with the data they need to inform effective scaling of the intervention in their states. Findings from the baseline phase of the study are presented in this paper and will be compared with follow-up data, to be collected at two future time points, following previous research on antenatal care interventions in low-and-middle-income countries (LMIC) [13].

### Study Objectives

This phase of the study sought to estimate baseline parameters on antenatal history and behavior, delivery preparedness, and perceptions of contraception among pregnant women attending gANC in health facilities in Kaduna and Kano States of Nigeria. It also elicited baseline data on the quality of ANC experienced by participants attending gANC in these health facilities.

## 2. Materials and Methods

### 2.1. Conceptual Model and Approach

This implementation research is underpinned by the Diffusion of Innovation theory [14,15]. The study explores how gANC is adopted and becomes a standard of care in Northern Nigeria [16]. This model provides a basis for the conceptual framework of the program, which posits that ANC utilization can be influenced to promote the adoption of change, which in this case is the gANC intervention [16]. Observing the implementation of gANC in real life can provide important insight into facilitators and barriers that should be considered in planning for the scaling of the intervention in the states. Figure 1 provides the theory of change, based on the Diffusion of Innovation theory, from the implementation of project activities to the observation of specific intermediate, primary, and long-term outcomes.

### 2.2. Study Design

The overall evaluation includes both quantitative and qualitative components, with the quantitative component being a three-wave longitudinal study of pregnant women enrolled in gANC at health facilities in Kaduna and Kano States. This paper focuses on the quantitative component of the overall study. Baseline data from that longitudinal study are the focus of the current report. Subsequent waves will include measures of uptake of, services received in, and satisfaction with care for gANC services over the course of the pregnancy, as well as perinatal outcomes including facility delivery, completion of postnatal health checks, postpartum family planning use, and uptake of recommended child immunizations.

### 2.3. Study Setting

Kaduna and Kano States are in the northwest geopolitical zone of Nigeria, with a projected 2022 population size of about 9.02 million and 15.46 million, respectively. Kaduna State has 23 local government areas (LGAs) spread across 3 senatorial districts in a land space of 45,061 km^2^, with a population density of over 200/km^2^. Kano State has a smaller land mass of about 20,230 km^2^ and a population density of 764.3/km^2^, with 44 LGAs in 3 senatorial districts. Nigeria’s 2006 census showed a gender distribution of 53% males and 47% females for Kano State, with about 50% of the population in the 15–64 years age category. Kaduna State had over 51% of the population within the age of 15–64 years, with a gender distribution of 51% males and 49% females. Islam is the predominant religion in both states, but with a higher proportion in Kano State. In Kaduna State, the total number of health facilities offering the gANC program is 486 and in Kano State it is 523.



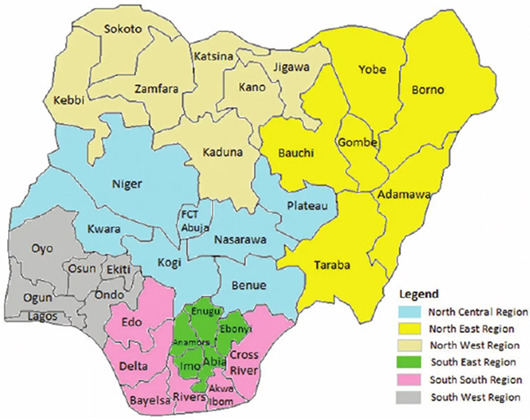



### 2.4. Study Population and Sample Size

Pregnant women enrolled in the gANC program in the participating health facilities in Kaduna and Kano States of Nigeria. In order to be eligible for the program, women needed to have visited one of the facilities offering gANC in Kaduna and Kano States that were selected for the study sample. We calculated that a sample of 1200 participants per state (2400 total) would be sufficient to provide statistical power of at least 0.80 in the planned propensity score analyses. The key assumptions in that calculation were the following: (a) that the marginal correlation between number of gANC sessions attended and the probability of facility delivery, including covariates used to obtain the propensity scores, will be 0.10; and (b) that no more than 30% of enrolled participants will be lost to follow-up between baseline and subsequent waves of the longitudinal study.

### 2.5. Sampling and Recruitment Strategy

Statistical power calculation was 1200 per state (2400 total), based upon the plan to conduct propensity score matching (PSM) analysis to isolate the cause-and-effect relationship between number of gANC meetings attended and the probability of delivering in a health facility. A stratified, three-stage cluster sampling design was used. In the first stage, the local government areas (LGAs) in each state were stratified by senatorial district, of which there are three in each state, and we randomly sampled two LGAs per senatorial district with equal probability, for a total sample of 12 LGAs. At this stage, some LGAs were removed from the sampling frame because the security situation would make data collection there unsafe for study personnel. In the second stage, we randomly sampled eight health facilities within each sampled LGA, again with equal probability, yielding a total of 96 sampled facilities. In the final stage, we planned to sample 25 eligible women per sampled facility. Some sampled facilities, however, were excluded because of security concerns, because they had insufficient enrollment, or because they were not implementing gANC at the time of baseline data collection. To compensate for this, we sampled more than 25 women from the remaining facilities in LGAs in which some facilities were excluded, in order to achieve a sample of 200 women in each LGA. Pregnant women aged 15–49 years attending gANC who consented to participate were recruited for the study.

### 2.6. Data Collection

Baseline data collection for this study was conducted in Kaduna and Kano States in March and April of 2024. This initial phase of the study included both qualitative and quantitative baseline components. A 72-item questionnaire was used for the quantitative survey. The survey tool was pilot-tested prior to implementation. Data collection was conducted during 15 days of fieldwork in March 2024. Eligible women (age 15–49 years at 12–23 weeks GA) were recruited from randomly selected sample of facilities in each state. In each state, the quantitative survey was composed of three broad teams of interviewers and supervisors, with one team per senatorial district. In each senatorial district, 10 interviewers and 2 supervisors conducted fieldwork.

### 2.7. Data Analysis

For the baseline data, initial data quality and consistency checks were conducted, and data cleaning was performed before analyses. Frequencies, descriptive statistics, cross-tabulations for the baseline data were developed overall and by state. This covered multiple domains such as sociodemographic background, prior pregnancies and births, current pregnancy and gANC use, services received, quality of care, and satisfaction with care. All analyses were carried out in Stata SE v18 (College Station, TX, USA).

## 3. Results

A total of 1269 and 1200 participants were recruited from the sampled facilities in the six sampled LGAs of Kaduna State and Kano State, respectively.

### 3.1. Sociodemographic Data of Participants

The sociodemographic composition of the sample is shown, overall and by state, in Table 1. Participants in both states were mostly between ages of 19 and 31 years, married or living with a partner, and identified as Muslim. Over 62% and 34% had completed secondary or higher-level education in Kaduna and Kano States, respectively. In both states, just over half of the participants have their own business. While nearly 60% of participants from Kaduna lived in urban areas, over 80% of participants from Kano were in rural areas. Only about 27% of the study participants in Kaduna State and 22% in Kano State had never gave birth before (Table 1). Up to 71% had 3 or more living children, with 10% having up to 6 or more children in Kaduna State. In Kano State, 61% had 3 or more living children, with about 20% having 6 or more children.

### 3.2. Obstetric History

#### 3.2.1. Previous Pregnancies

Table 2 summarizes information about prior pregnancies among the subset of participants who reported having at least one prior birth. In Kaduna State, among those who reported at least one previous pregnancy and birth, nearly 65% of participants’ last delivery was in a facility, over 70% had skilled birth assistance, and nearly 70% reported no problems with previous pregnancies. About 88% had postnatal care appointments, and 83% had a postnatal check within one week of delivery. In Kano, 61% of participants with at least one prior pregnancy and birth reported that their last delivery was at home, up to 42% had no skilled birth assistance, while 64% reported no problems with previous pregnancies. Up to 82% had postnatal care appointments, and 80% had a postnatal check within one week of delivery.

#### 3.2.2. Current Pregnancies

As shown in Table 3, in both states, about two-thirds of the participants had a first ANC visit by 18 weeks gestational age and had at least two visits.

### 3.3. Quality of Care

When asked about their most recent ANC visit, in both states, over 90% of participants reported receiving at least 4 services, and over 75% reported receiving no less than 9. Discussing iron folic acid (IFA) side effects with the provider was the least common service received, at just over 20%. Other services not commonly provided included estimation of delivery dates, height measurement, and asking about blood group (see Appendix A for findings). In both states, nearly all participants expressed satisfaction with the antenatal care they received with over 95% planning to continue ANC at the same facility.

### 3.4. Measures of Birth Readiness

In Kaduna State, nearly 90% of participants had identified a delivery facility, and among those, over 70% had made a transport plan. Over 35% of participants reported it took 30 min or more to reach a facility. Nearly 70% of participants reported having a cash plan for emergency care, and 90% had made arrangements for a companion. In Kano, over 50% of participants had identified a delivery facility, and among those, nearly 80% had made a transport plan. A third of the participants reported it took 30 min or more to reach a facility. Nearly 80% of the participants reported having a cash plan for emergency care, and nearly all had arrangements for a companion (Appendix A). More than 85% of participants plan to be at home during pregnancy and after birth.

### 3.5. Family Planning Behavior

About half of participants in Kaduna State had not previously used family planning methods, and over three-quarters of those who had used them said it was a joint decision between them and their husbands/partners. More than half reported they could ask their partner to use a condom, over 90% said they would delay future pregnancy and said there were advantages to spacing births. In Kano State, over 60% of the participants had previously used family planning, and 65% said it was mainly a joint decision. Nearly half reported they could not ask their partner to use a condom, over 3/4 said they would delay future pregnancy, and over 90% said there were advantages to spacing births.

### 3.6. Pregnancy Knowledge and Behavior by Education Status

Across education levels, there was agreement that spacing pregnancies is advantageous. This perception tended to be higher among participants with higher education levels. Reported ability to ask for condom use was also higher among participants with higher education levels (Appendix A). There are high levels of awareness of warning signs, and this also tends to be higher among participants with higher education levels. Most participants had made transport plans, and this was slightly higher among participants with higher education levels. Most participants across education levels had had 1–3 ANC visits (Appendix A). Plans to deliver at a facility were higher among participants with higher education levels (Figure 2).

## 4. Discussion

This study provides baseline data for a multi-year evaluation process of the implementation of the gANC program in Nigeria. Although inferences about the impacts of gANC utilization cannot be made from the data at this phase, some important observations from baseline are noted in comparison with data from the literature. The experience and findings from this initial implementation evaluation provide context and insights for the planned longitudinal follow-up and data analysis for this project.

In both states, over 95% of the participants expressed satisfaction with the individual antenatal care (ANC) they had received in previous pregnancies from the same facility. Considering that gANC is a novel approach and most participants had received minimal doses at baseline, it is not surprising that they consider the status quo standard and satisfactory, not having a higher quality experience to compare with. Social desirability bias is also not unlikely, where participants inaccurately provide a desirable response to maintain a positive impression and not displease the interviewer [17]. High level of satisfaction with ANC had been observed by other studies in this region. In a survey of 1336 mothers in Northern Nigeria, 90% of the respondents expressed satisfaction with the ANC services and this was associated with responsive treatment, communication, staff being empathetic, ease of access to treatment, and other quality-related factors [18]. Triangulating qualitative data in this study could provide more insight into the perception of satisfaction and related factors and provide basis for subsequent comparisons.

Over 40% of the participants in Kano State had their last delivery at home with no skilled birth attendant. This is not an unusual finding in this part of Nigeria. This seems to contradict the high response of satisfaction expressed in ANC. It is possible that experience at ANC differs from that during childbirth in health facilities. This was the observation in an earlier study in Kano State where women quit ANC to deliver at home due to the attitude of maternity staff and the presence of male staff during deliveries [15]. Facility-based delivery was observed to be higher among women who participated in gANC compared to their ANC counterparts in a facility-based, pragmatic, cluster-randomized controlled trial of Nigerian women in Nasarawa state, Northcentral Nigeria [7]. The quality of antenatal care received in gANC could have been responsible for this positive effect in the choice of health facility delivery. In another study, a strong correlation was found between ANC attendance and the use of skilled birth attendant (r = 0.706, *p* < 0.001) in a secondary analysis of the 2018 Nigerian National Nutrition and Health Survey, further confirming that antenatal care experience could influence delivery behavior choices [19]. The positive effect of gANC on delivery behavior choices is anticipated in this present implementation study that has facility-based delivery as a primary outcome.

One in four participants in Kano State commenced ANC at ≥23 weeks gestational age. This is similar to the findings from the national survey, where up to 63% of women started ANC during the second trimester [3]. In Kaduna, women were more likely to commence ANC during the first trimester. Timely initiation of ANC has been associated with better health outcomes among pregnant women in Sub-Saharan Africa [20]. It would be valuable to explore factors that influence decision on when to commence ANC as this could guide targeted interventions to improve health outcomes among pregnant women in Nigeria.

The level of birth preparedness among women in both states was generally high, with most having identified delivery health facility, made transportation and cash plans for emergency, and identified a companion. Preparedness was reported more by those with higher levels of education, including intention to deliver in a health facility. This is not surprising, as education is known to influence health literacy, which in turn leads to adoption of health-promoting behavior [21,22]. This was also the case in the observed higher proportions of those with higher levels of education perceiving child spacing as advantageous and being able to ask their husbands/partners to use condom. Up to half the participants in Kano State expressed their inability to ask their husbands/partners to use condom. This is similar to the findings from a study on the use of condom by women in Botswana, where women were less likely to use condoms if they perceived their partners would be angry [5]. Interventions that improve health literacy are therefore recommended for better reproductive health behavior and outcomes. This has been proposed as a life-course strategy that can be implemented in many settings, including informal learning settings [4]. It is hoped that the peer-to-peer learning from gANC will support health literacy and improve pregnancy outcomes among the participants.

### 4.1. Limitations

This is a baseline study and report on the methodology used in the gANC implementation evaluation. As such, we cannot draw conclusions at this point about the impact of the program. Future longitudinal data collection using PSM analysis to create a matched-comparison group will enable us to make inferences about program effectiveness. While PSM is a sound quasi-experimental methodology, and the best available analytic method given that the gANC program was being implemented in Kaduna and Kano before the evaluation began, it is not a randomized controlled design. The quality of the PSM method will depend on having sufficient matching data, and on potential loss to follow-up in the longitudinal sample, which will be determined in the next phase of the project.

### 4.2. Future Research

We will continue with midline data collection in late 2024, and endline data collection in 2025. Longitudinal data collection and efforts to maximize participant retention and minimize loss to follow-up will be the key next steps. Once follow up data are available, we will use PSM to create comparison groups (treatment and comparison) based on matching of demographic characteristics. The PSM analysis will be based on several sociodemographic variables, including location, gender, religion, education, employment, and other covariates, in a multivariable logistic regression procedure to compare participants who received more gANC services (attendance at meetings) during the implementation period to those who participated in fewer or no intervention services [23]. This procedure will be used to create two matched groups for subsequent multivariate impact analysis where the independent variables will be participation in gANC meetings, and the dependent variables will be giving birth at a facility (primary) as well as additional secondary outcomes [24].

Future studies should include evaluating gANC in other settings within Nigeria and other low- and middle-income countries that have low utilization of ANC in healthcare facilities. Studies should examine approaches to improving the quality of care following the WHO guidelines [11]. Such studies may examine programs designed to reduce the risk of stillbirths and pregnancy complications and give women a positive pregnancy experience, following WHO recommendations. Use of novel methods to promote ANC utilization, including reducing barriers such as transportation and cost, and exploring telehealth solutions and mobile health technologies should also be examined.

## 5. Conclusions

An evidence-based gANC program is being implemented across two states in Northern Nigeria. An implementation evaluation has been successfully launched and has collected data from 2469 women receiving care at facilities delivering the gANC program in Kaduna and Kano States. Baseline data shows high level of satisfaction with ANC services and high level of birth preparedness by women in both states. However, most women in Kano State commenced ANC after the fifth month of pregnancy and had high rate of home deliveries with unskilled birth attendants in their last pregnancy. Close to 50% of pregnant women in Kaduna State had not used contraceptives before. Longitudinal data collection will continue at two future time points and utilize a PSM analysis methodology to identify potential impacts of the gANC program.

## Figures and Tables

**Figure 1 ijerph-21-01461-f001:**
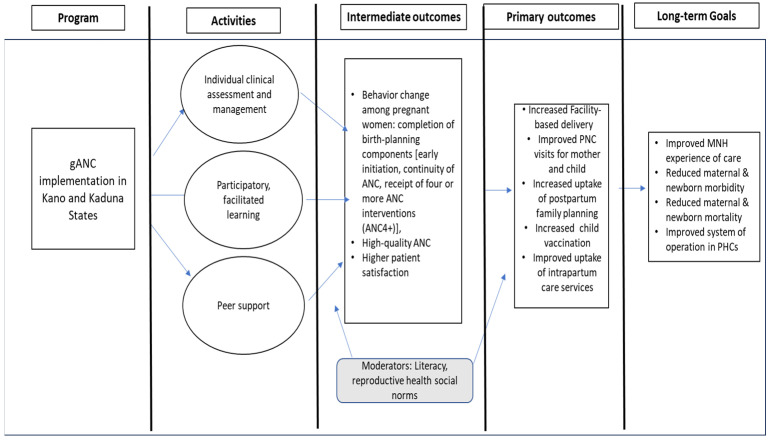
Theory of change for gANC.

**Figure 2 ijerph-21-01461-f002:**
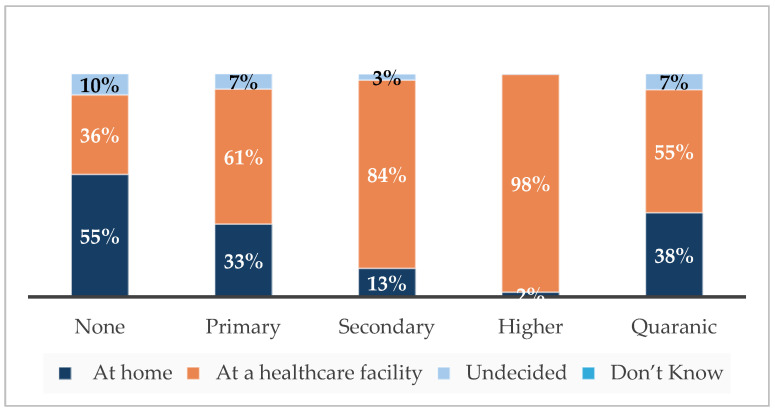
Intended delivery site by education level.

**Table 1 ijerph-21-01461-t001:** Sociodemographic characteristics and past obstetric history of the sample, by state.

Variable	Kaduna (n = 1269)	Kano (n = 1200)	Total (n = 2469)
Age (M(SD))	25.2 (5.7)	25.5 (5.9)	25.3 (5.8)
Marital Status			
Married or living together	98.2%	99.1%	98.6%
Divorced/separated	0.4%	0.8%	0.6%
Widowed	0.2%	0.2%	0.2%
Never married and never living together	1.2%	0.0%	0.6%
Religion			
Catholic	7.4%	0.2%	3.9%
Other Christian	14.0%	0.2%	7.3%
Islam	78.6%	99.5%	88.7%
Traditionalist	0.0%	0.1%	0.0%
Other	0.0%	0.0%	0.0%
Education			
Never attended formal school	11.9%	35.4%	23.3%
Primary	18.8%	23.4%	21.0%
Secondary	44.7%	29.2%	37.3%
Higher	17.7%	5.2%	11.6%
Quranic/Islamiyya	6.8%	6.8%	6.8%
Employment Status			
Unemployed	40.6%	44.1%	42.3%
Employed full time	4.9%	1.4%	3.2%
Employed part time	4.0%	1.4%	2.8%
Employed own a business	50.5%	53.1%	51.8%
Residence			
Urban	58.0%	18.8%	38.9%
Rural	42.0%	81.2%	61.1%
Ever given birth before			
Yes	72.6%	78.3%	75.4%
No	27.4%	21.7%	24.6%
Number of Living Children			
0	27.4%	21.7%	24.6%
1	1.8%	1.5%	1.7%
2	22.9%	16.1%	19.6%
3	15.9%	15.4%	15.7%
4	13.0%	14.3%	13.6%
5	8.9%	11.2%	10.0%
6+	10.0%	19.8%	14.8%

**Table 2 ijerph-21-01461-t002:** History of previous pregnancies by state.

Previous Pregnancies	Kaduna (n = 960)	Kano (n = 990)	Total (n = 1950)
Time since last birth (M(SD))			
Problems during previous pregnancies			
Yes	30.4%	36.1%	33.3%
No	69.5%	63.7%	66.6%
Do not know	0.1%	0.2%	0.2%
Location of last delivery			
At home	34.3%	60.7%	47.7%
In a healthcare facility	63.5%	36.3%	49.7%
In transit to a healthcare facility	0.6%	0.4%	0.5%
Other	1.6%	2.6%	2.1%
Skilled birth attendant present			
Yes	71.8%	57.8%	64.7%
No	26.7%	41.6%	34.3%
Do not know	1.6%	0.6%	1.1%
Postnatal health check within one week			
Yes	83.2%	79.7%	81.4%
No	15.9%	20.1%	18.1%
Do not know	0.8%	0.2%	0.5%
Any postnatal care appointments			
Yes	88.2%	82.3%	85.2%
No	10.5%	17.5%	14.1%
Do not know	1.2%	0.2%	0.7%

**Table 3 ijerph-21-01461-t003:** Current pregnancies and ANC uptake.

Variable	Kaduna (n = 1269)	Kano (n = 1200)	Total (n = 2469)
Gestational age (M(SD))	19.6 (5.8)	20.9 (5.9)	20.2 (5.9)
Gestational age at first ANC visit			
12–15 weeks	39.3%	20.2%	30.0%
16–18 weeks	27.7%	23.7%	25.7%
19–20 weeks	15.8%	23.0%	19.3%
21–22 weeks	4.5%	7.6%	6.0%
23 weeks or later	12.5%	25.2%	18.7%
Do not know	0.2%	0.4%	0.3%
Number of ANC visits			
0	0.1%	0.0%	0.0%
1	35.4%	32.5%	34.0%
2	40.7%	33.9%	37.4%
3	13.8%	17.8%	15.7%
4	5.8%	8.9%	7.3%
5	2.6%	4.9%	3.7%
6	1.1%	1.3%	1.2%
7	0.5%	0.2%	0.4%
8	0.1%	0.2%	0.2%
9	0.0%	0.1%	0.0%
10	0.0%	0.1%	0.0%
Identified a facility for delivery			
Yes	87.5%	53.2%	70.8%
No	10.6%	39.3%	24.6%
Do not know	1.8%	7.5%	4.6%

## Data Availability

Dataset available on request from the authors.

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
