# Peer review of "Implementation and Baseline Evaluation of an Evidence-Based Group Antenatal Care Program in Two Nigerian States"

_ijerph, 2024, doi:10.3390/ijerph21111461_

Round 1

Reviewer 1 Report

Comments and Suggestions for Authors

The manuscript ‘Implementation and Evaluation of an evidence-based group antenatal care program in two Nigerian states: Design and baseline data collection’ is an interesting read. The authors have described the study design and baseline characteristics of the study participants of a group antenatal care program in two states in Nigeria.

There are a few points to raise.

Abstract

Line 17: The authors state that ‘Group antenatal care (gANC) improves ANC behavior and pregnancy outcomes’. This sounds like a conclusion has already been drawn even before carrying out their study.  The authors may want to rewrite the sentence to read ‘Previous studies have shown that group antenatal care (gANC) improves ANC behavior and pregnancy outcomes’

Introduction

Line 38: please insert space between ‘33.8%’ and ‘compared’

Lines 44 - 45: ‘Group ANC is a concept that involves enable peer-to-peer discussions and sharing of experiences’ This sentence is incomplete and needs to be rewritten for clarity

Line 81: What is BOWs?

Lines 84 – 86: can be rewritten to read ‘Findings from the baseline phase of the study are presented in this paper and will be  compared with follow up data to be collected at two future time points’

Materials and Methods

Line 121: please insert space between ‘Nigeria’s’ and ‘2006’

Line 126: ‘Kano State’ not ‘Kano Stater’

Line 140: PSM should be written in full as it is used for the first time in the manuscript

Line 161: please insert space between ‘March’ and ‘2024’

Lines 161 – 162: how was the gestational age verified to ensure that those enrolled were actually between 12 and 20 weeks? Was it based on reported gestational age? This should be clearly stated

Results

Lines 181 – 182: ‘Nearly 60% and over 80% live in an urban area in Kaduna and Kano States respectively’ This statement differs from what is presented in Table 1 about Kano State. The percentages for Kano State are reversed in Table 1

Line 184: please insert space between ‘to’ and ‘6’

Table 1: total is 2469 not 1469

Line 202: please insert space between ‘by’ and ‘18’

Table 3: Table 1: total is 2469 not 1469

Table 3: The mean GA is 19.6 for Kaduna and 20.9 for Kano, this is impossible if the GA range was 12- 20 weeks as stated in lines 161 – 162.

Also Table 3 shows that the GA at first ANC visit was ‘21-22 weeks’ or ‘23 weeks or later’ in some of these women. How is this possible if they were supposed to be between 12 and 20 weeks GA? This needs to be properly clarified

Line 227: there is a missing information between ‘and’ and ‘said’

Discussion

Lines 284 to 286: can be rewritten to read ‘It would be good to explore factors that influence decision on when to commence ANC as this could guide targeted interventions to improve health outcomes among pregnant women in Nigeria’

Comments on the Quality of English Language

The quality of English Language is excellent, there are just a few errors which have been pointed out

Reviewer 2 Report

Comments and Suggestions for Authors

Article

Implementation and Evaluation of an evidence-based group antenatal care program in two Nigerian states: Design and baseline data collection

Thank you for the opportunity to review this article, which covers a central topic within perinatal care.

The article is a well-structured contribution to the literature on maternal health in low-resource settings. The authors provide a solid foundation with baseline data, though future papers would benefit from deeper analysis and the inclusion of qualitative data to provide a fuller picture of gANC’s impact. As it stands, the study offers valuable insights into the implementation of gANC in Northern Nigeria and has the potential to inform policy and practice significantly.

However, I would like to highlight some observations that could enhance the article. I will proceed in order:

It is recommended to uniform the font and font size throughout the entire text, particularly in the section of affiliations to maintain consistency across the article.

Title

Beginning with the title, I propose a revision to enhance its conciseness and compactness: Implementation and Baseline Evaluation of an Evidence-Based Group Antenatal Care Program in Two Nigerian States.

Abstract:

The abstract could be improved to enhance the study's effectiveness. Some sentences could be simplified, for example, line 18-20 could be revised, as follows: This paper presents baseline findings from the implementation of the group antenatal care (gANC) program in Kaduna and Kano states, Nigeria, based on data collected from 1,269 and 1,200 pregnant women, respectively.

Additionally, it currently lacks data on the results obtained thus far. It includes specific demographic characteristics that may not be suitable for an abstract. It would be beneficial to conclude with a statement highlighting future perspectives in antenatal care (ANC) or emphasizing the study's significance for health policy.

Introduction:

The introduction seems well-argued with several studies highlight Group Antenatal Care (gANC) as a promising intervention to enhance antenatal care (ANC) uptake, improve the quality of care, and positively impact maternal outcomes, particularly in low- and middle-income countries. By drawing on research findings from various settings, the argument convincingly demonstrates that gANC can address the challenges faced by two states, such a living distance to health facility, healthcare experience, gender relations and contraceptive uptake. Otherwise, i propose simplifying and condensing lines 34 to 37 to improve readability, as follows: "Nigeria has historically low rates of antenatal care (ANC) utilization, with pronounced disparities in rural areas and the northwest region."

Methods: The use of a longitudinal study design with a mixed-methods approach offers valuable insights. The combination of quantitative and qualitative data is well-suited for implementation evaluation, and the use of the Diffusion of Innovation theory provides a solid conceptual framework for understanding the adoption of gANC.

Discussion:

Overly technical phrases should be simplified to enhance accessibility. For example, the phrase "This study has provided baseline data in the first step in a multi-year process" can be shortened to "This study provides baseline data for a multi-year evaluation process."
Additionally, the discussion could delve deeper into the implications of differences in socio-demographic characteristics between Kaduna and Kano, particularly regarding urban-rural disparities and their potential effects on gANC utilization and maternal health outcomes.

References:

It is recommended to ensure that all references are consistent. Some inconsistencies, such as spacing between authors, should be corrected.

Comments on the Quality of English Language

The Quality of English Language is okay.

Reviewer 3 Report

Comments and Suggestions for Authors

The authors have presented an important study of "Implementation and evaluation of an evidence-based group-based antenatal care programme in two states of Nigeria: design and collection of baseline data" with special relevance to the region as a reference.

Just to clarify:

1. Which non-statistical variables influenced the results obtained. For example: cultural level, religion, pregnancy history, etc.

2. What is the strategy for disseminating these results to the Nigerian community and becoming a reference for the evaluation of the gANC programme

Reviewer 4 Report

Comments and Suggestions for Authors

This manuscript presents initial results from a study that seeks to demonstrate the effectiveness of Antenatal Care/Postnatal Care in Nigeria, particularly a group antenatal care program in 2 states of Nigeria.

The paper is well written and presents interesting findings. As stated by the authors: "The experience and findings from this initial implementation evaluation provide context and insights for the planned longitudinal follow-up and data analysis for this project."

I have some points to be addressed to improve the quality of the manuscript:

1. A conclusion should be added to the abstract.

2. Title of the section 2.1 is formatted incorrectly.

3. Consider "This implementation research is underpinned by the Diffusion of Innovation theory..." and "Figure 1 provides the theory of change..."; Are the "Diffusion of Innovation theory" and the "theory of change" the same? The text is confusing. Unpack this explanation. Figure 1 could be explained step by step.

4. A map could benefit section 2.3. Study setting. Also, data provided in a table could be better, because differences and similarities could be more evident (i.e., easy to compare the states).

5. I could not get the point of this '(a)': "Key assumptions in that calculation were (a) that the..."

6. Provide details for 'Statistical power calculation'. How did you calculate it?

7. Consider "Pregnant women aged 15-49 years..."; why was 'age' used as a selection criterion?

Reviewer 5 Report

Comments and Suggestions for Authors

Please find below my comments:

1. The title is clear

2. The abstract represents the study, however, the conclusion in the abstract should be properly stated.

3.The introduction part is well-written and provide the study rationale very clearly.

4. The methods section provides a detailed description of the methods utilized, ensuring the study's reproducibility.

5. The results part is interesting and is supported by clear and high-quality visual materials.

The discussion part is narrow and does not properly discuss the received results.

Suggest rewriting and restructuring it using the following outline:

1.1 Rationale of the study (why it was done)

1.1.1 Main findings of the study

1.1.2 What makes your study unique

1.1.3 What it adds to what we already know

1.2 Study subjects

1.3 Subject of the discussion

Comparison of your results with neighboring countries, with countries of the same development levels (income), with developed high-income countries). Agreement and disagreement with the studies compared. relevant studies for comparison - DOI: 10.1097/XEB.0000000000000234; doi: 10.1097/XEB.0000000000000230; 

1.4 Summ up of the study, study strengths and limitations

1.5 Clinical implication 

6. The conclusion should be expanded to reflect the study findings.
